# As the Pandemic Progresses, How Does Willingness to Vaccinate against COVID-19 Evolve?

**DOI:** 10.3390/ijerph18020797

**Published:** 2021-01-19

**Authors:** Stephanie J. Alley, Robert Stanton, Matthew Browne, Quyen G. To, Saman Khalesi, Susan L. Williams, Tanya L. Thwaite, Andrew S. Fenning, Corneel Vandelanotte

**Affiliations:** 1School of Health, Medical and Applied Sciences, Building 7, Central Queensland University, Bruce Highway, Rockhampton, QLD 4702, Australia; r.stanton@cqu.edu.au (R.S.); q.to@cqu.edu.au (Q.G.T.); s.khalesi@cqu.edu.au (S.K.); s.p.williams@cqu.edu.au (S.L.W.); t.thwaite@cqu.edu.au (T.L.T.); a.fenning@cqu.edu.au (A.S.F.); 2Physical Activity Research Group, Appleton Institute, Central Queensland University, 44 Greenhill Road, Wayville, SA 5043, Australia; 3Cluster for Resilience and Wellbeing, Appleton Institute, Central Queensland University, 44 Greenhill Road, Wayville, SA 5043, Australia; 4School of Health, Medical and Applied Sciences, Central Queensland University, 6 University Drive, Branyan, QLD 4670, Australia; m.browne@cqu.edu.au

**Keywords:** COVID, coronavirus, hesitancy, acceptance, time, demographics

## Abstract

Controversy around the safety and efficacy of COVID-19 vaccines may lead to low vaccination rates. Survey data were collected in April and August 2020 from a total of 2343 Australian adults. A quarter (*n* = 575, 24%) completed both surveys. A generalized linear mixed model analysis was conducted to determine whether willingness to vaccinate changed in the repeated sample, and a multinominal logistic regression was conducted in all participants to determine whether willingness to vaccinate was associated with demographics, chronic disease, or media use. Willingness to vaccinate slightly decreased between April (87%) and August (85%) but this was not significant. Willingness to vaccinate was lower in people with a certificate or diploma (79%) compared to those with a Bachelor degree (87%), *p* < 0.01 and lower in infrequent users of traditional media (78%) compared to frequent users of traditional media (89%), *p* < 0.001. Women were more likely to be unsure if they would be willing to vaccinate (10%) compared to men (7%), *p* < 0.01. There were no associations between willingness to vaccinate and age, chronic disease, or social media use. Promotion of a COVID-19 vaccine should consider targeting women, and people with a certificate or diploma, via non-traditional media channels.

## 1. Introduction

The spread of the novel coronavirus (COVID-19) has resulted in a worldwide pandemic. In the last year since the original outbreak in December 2019 there have been over 80 million confirmed cases, and over 1.8 million associated deaths worldwide [1]. In Australia there have been over 28,000 confirmed cases and more than 900 associated deaths [2]. Numerous medical and research institutions have developed vaccines for COVID-19 [3]. Clinical trials support the safety and efficacy of vaccines developed by Oxford-AstraZeneca and Pfizer and the Australian government have registered their intention to purchase these vaccines [3]. The United States and United Kingdom have recently started to roll out the Pfizer vaccine and preliminary findings suggest efficacy of the vaccine to be as high as 95% [4]. For a COVID-19 vaccine with 80% efficacy, it is estimated that at least 60% of the population will need to receive the vaccination to achieve herd immunity [5].

Around 95% of 5-year olds in Australia have received all routine childhood vaccinations [6]. However, uptake of the voluntary seasonal influenza vaccination is much lower at 52% [7]. Due to controversy around the perceived rapid development, safety, efficacy and forced delivery of a COVID-19 vaccine, uptake may also be lower than that of routine childhood vaccinations. A survey conducted during the initial wave of the pandemic in Australia (April 2020) found that 86% of Australians were willing to receive a COVID-19 vaccine (another 9% were unsure, and 5% were unwilling) [8]. However, some evidence suggests that willingness to vaccinate against COVID-19 is changing in Australia [9]. The survey was conducted again in June and July 2020 when COVID-19 transmission had slowed and social restrictions were easing. Results showed a slight increase in willingness to vaccinate where 87% of June participants and 90% of July participants were willing to receive a COVID-19 vaccine [10]. Conversely, a study conducted in Italy found willingness to vaccinate against COVID-19 decreased over time and as COVID-19 transmission rates decreased [11]. Another Australian survey conducted in June 2020 found a lower proportion of Australians were willing to receive a COVID-19 vaccine (75% with 17% unsure, and 8% unwilling) [9]. However, this study only included parents recruited through the Royal Children’s Hospital, Melbourne. A limitation of the available data is a lack of longitudinal study designs, hence different findings at different timepoints may be due to differences in participants, rather than a true change in willingness to vaccinate against COVID-19. It is therefore unknown if there is a decreasing trend of willingness to vaccinate against COVID-19 in Australia.

No strong or consistent associations have been found between attitudes towards routine vaccines and demographics such as age, gender, education, occupation, and income [12,13]. However, some limited research in Australia has demonstrated that males, older adults, and people with a higher education are more likely to be willing to receive a COVID-19 vaccination [8,9]. There is evidence that good physical health is associated with attitudes towards routine childhood vaccines and willingness to receive the influenza vaccine [14,15]. It is likely that health status may also be associated with willingness to vaccinate against COVID-19, as individuals with pre-existing health conditions such as diabetes, severe asthma, hypertension and cardiovascular diseases are more likely to experience severe health problems if they contract COVID-19 [16].

People who frequently access news may be more aware of the impact of the COVID-19 pandemic, both in their communities and worldwide, and be more informed of COVID-19 vaccine development and clinical trials. This may differ, however, when news is accessed via traditional media sources such as television or newspapers versus non-traditional sources including the internet and social media, the latter of which has higher rates of fake news and non-evidence-based health information [17,18,19]. Disinformation regarding the safety and efficacy of vaccinations is rife on social media and the internet generally [20]. Consistent with this, use of the internet, particularly social media for obtaining health information, is associated with negative attitudes towards routine vaccines [13,19]. Accordingly, the degree to which individuals obtain information from non-traditional sources, rather than traditional sources, may present an important explanatory factor in public willingness to receive the COVID-19 vaccine.

As only limited studies have investigated changes in willingness to vaccinate against COVID-19 over the course of the pandemic, or investigated whether health status and media use are associated with willingness to vaccinate against COVID-19, the main aim of this study is to determine whether willingness to vaccinate against COVID-19 changed from the beginning of COVID-19 pandemic (April 2020) to four months later (August 2020). The second aim is to determine whether demographic factors, chronic disease status, traditional media use and social media use are associated with willingness to vaccinate. Findings will be valuable for the targeted promotion of a COVID-19 vaccination once it becomes available.

## 2. Materials and Methods

### 2.1. Participants and Recruitment

Participants were recruited to complete an online survey on Qualtrics. Qualtrics is an online platform which hosts secure surveys and is based in Seattle, Washington and Provo, Utah. Participants are directed to the survey through a direct link [21]. Participants were recruited through email lists, traditional news media (radio and television), social media networks and paid Facebook advertisements. The survey was distributed twice, the first iteration open between 9 and 19 April 2020, and the second between 30th July and 16 August 2020. Respondents to the first survey were asked to provide their email if they were interested in completing the second survey. For the second survey, new participants were also recruited through the same methods. Participants were eligible if they were Australian residents aged 18 years or over. Ethical approval was received from the Central Queensland University’s Human Research Ethics Committee (Approval number 22332) and the study was conducted according to the Declaration of Helsinki. Informed consent was obtained from all participants after they were shown the information sheet detailing the nature of participation and before they were given access to the survey.

The first survey was distributed early in the pandemic when COVID-19 transmission rates were increasing throughout Australia [2]. There were Australia-wide isolation laws where residents were only permitted to leave their home to buy essential items such as groceries, obtaining medical care, essential work, or exercising. Interstate travel was banned, and travel within states was restricted. Schools were closed and universities ceased face-to-face teaching. People were encouraged to work from home if possible and public gatherings were banned. There was a limit of five people at private gatherings, including weddings and funerals [22]. During distribution of the second survey, COVID-19 transmission rates had decreased throughout Australia (with the exception of the state of Victoria where COVID-19 transmission rates were increasing [2]). In response to the decline in transmission rates, Australia’s isolation laws in all states, except Victoria, had reduced significantly and residents could travel freely within states and between some (but not all) states. Schools and many workplaces had reopened, and public gatherings were allowed if a space of four square meters per person was available [22]. Conversely, strong lockdown laws in Victoria were reinstated a few days after distribution of the second survey began. These new lockdown laws were stricter than those during the first wave and survey, as they also included closure of childcare centers, no private gatherings, travel only within 5 km of home, and an 8 pm to 5 am curfew [23].

### 2.2. Materials

The survey included questions on demographics, chronic disease status, media use, personal experience with COVID-19, resilience, mental health, compliance with COVID-19 government recommendations, health behaviours, and willingness to vaccinate against COVID-19. Further details of the survey can be found elsewhere [24]. Only the questions on demographics, chronic disease status, media use, and willingness to vaccinate against COVID-19 are included in the current paper (see Appendix A for complete questions).

Demographic data included age in years (categorised into 18–44, 45–64, >65), gender (male, female), education (year 12 or below, technical studies/certificate/diploma, bachelor’s degree or above), household income (<AUD$1000/week, 1000-<AUD$2000/week, ≥AUD$2000/week) and location (Australian state or territory of residence). Chronic disease status was assessed by asking respondents ‘Have you ever been told by a doctor that you have any chronic health problems?’ with response options of ‘yes’ and ‘no.’

Use of social media and use of traditional media was assessed by asking participants ‘How much do you currently use social media (e.g., Facebook, Reddit, Twitter)?’ and ‘How much do you currently listen to/read/watch news on the radio, newspaper or TV?’ with eight response options: (1) ‘never’ (2) ‘rarely’ (3) ‘< 1 h per day’ (4) ‘1–2 h/day’ (5) ‘3–4 h /day’ (6) ‘5–6 h /day’ (7) ‘7–8 h /day’ (8) ‘most of the day.’ These were collapsed into three categories: (1) <1 h /day (2) 1–2 h /day (3) ≥3 h/day to even the spread of participants in each category and allow for comparison.

Willingness to vaccinate against COVID-19 was assessed by asking participants to agree or disagree with: ‘If a new vaccine for COVID-19 was released that was proven to be safe and effective, I would get vaccinated immediately.’ Participants were also asked to respond to: ‘If a new vaccine for COVID-19 was released but had not yet been proven to be safe and effective, I would get vaccinated immediately’, and ‘Even before the COVID-19 pandemic, I have always vaccinated myself against diseases when recommended by health professionals.’ Response options were on a 5-point Likert scale ranging from strongly disagree (1) to strongly agree (5). Categories for willingness to vaccinate were collapsed into ‘agree,’ ‘neither agree nor disagree’ and ‘disagree’ to aid interpretation of the analyses. A fourth vaccination variable was created to determine the percentage of participants who were more likely or less likely to vaccinate against COVID-19 compared to other vaccines recommended by their doctor (more likely, equally likely, less likely). This was done by combining the two questions above which assessed (1) willingness to receive COVID-19 vaccine if safe and effective and (2) usual vaccine behavior.

### 2.3. Data Analysis

A longitudinal analysis was conducted in participants who completed both surveys (*n* = 575, 24%) to determine whether there were within person changes over time on willingness to vaccinate, and whether this differed by living location (Victoria, other Australian state or territory). First, descriptive statistics were reported for the repeated sample. Number and percentage were presented for each demographic variable for survey 1, willingness to vaccinate against COVID-19 for survey 1 and 2 and living location for survey 2. Secondly, descriptive statistics on willingness to vaccinate against COVID-19 (number and percentage) were presented by time and living location. Thirdly, a generalised linear mixed model analysis was conducted with a multinomial distribution and logit link. The outcome was willingness to vaccinate against COVID-19 with the categories ‘agree’, ‘neither agree nor disagree’ and ‘disagree’ (reference). The fixed effects were time (survey 1, survey 1) and living location (Victoria, other Australian state or territory). Subject was included as a random effect. Main effects for time and living location were analysed. Due to the contrasting lockdown conditions between the state of Victoria and the rest of Australia during the second survey, an interaction between time and living location was also included in the analysis.

A multinomial logistic regression was conducted on repeated cross-sectional data to test whether time, demographic factors, chronic disease status, traditional media use, and social media use were associated with willingness to vaccinate. First, descriptive statistics (number and percentage for each category) were reported for each variable for survey 1, survey 2, and both surveys combined. Participants who completed both surveys were excluded from the reporting and analysis of survey 2 to avoid correlated data that violates the assumption of independence. Secondly, descriptive statistics (number and percentage for each category) for willingness to vaccinate against COVID-19 were presented by time, demographics, chronic disease status, traditional media use, and social media use. Thirdly, a multinomial logistic regression was conducted with main effects for time (survey 1, survey 2), living location (Victoria, other state or territory), age (18–44, 45–64, >65), gender (male, female), education (year 12 or below, technical studies/certificate/diploma, Bachelor degree or above), chronic disease status (with, without), social media use (<1 h/day, 1–2 h/day, ≥3 h/day) and traditional media use (<1 h/day, 1–2 h/day, ≥3 h/day). An interaction between time and living location was also included in the analysis. The outcome was willingness to receive a COVID-19 if proven safe and effective with the categories ‘agree’, ‘neither agree nor disagree’ and ‘disagree’ (reference). Income was not included in the analysis due to a high number of missing values (*n* = 349, 15%). For comparison, the multinominal logistic regression analysis was repeated with the outcome of usual willingness to receive vaccines recommended by a doctor with the categories ‘agree’, ‘neither agree nor disagree’ and ‘disagree’ (reference). This secondary analysis is reported in Appendix A.

## 3. Results

### 3.1. Longitudinal Analysis in Participants Who Completed Both Surveys

Table 1 presents the descriptive statistics of demographics and willingness to vaccinate against COVID-19 in the sample of participants who completed both surveys. The repeat sample were mostly over 55 years of age (51%), female (70%), had a Bachelor degree or above (63%) and had a household income of over $2000 per week (41%). About half had a chronic disease (52%) and less than a quarter were Victorian residents (19%). Willingness to vaccinate slightly decreased between time 1 (87%) and time 2 (85%).

The longitudinal analysis (Table 2 and Table 3) of participants who completed both surveys demonstrated that 88% of non-Victorians at time 1 and 84% at time 2 were willing to vaccinate against COVID-19 (−4%). In comparison, 85% of Victorians at time 1 and 89% at time 2 were willing to vaccinate against COVID-19 (+4%). The generalised linear mixed model showed a slight overall decrease in willingness to vaccinate within participants. Although it did not meet criteria for statistical significance (*p* = 0.06), there was some indication that Victorians were more likely to increase their willingness to vaccinate between time 1 and 2.

### 3.2. Repeated Cross-Sectional Analysis in All Participants

Table 4 presents descriptive statistics for demographic characteristics, chronic disease status, media use, and willingness to vaccinate against COVID-19 for survey 1, survey 2, and combined surveys. Characteristics of respondents to both surveys combined (*n* = 2343) indicate that half of the sample was over 55 years old (46%). A higher percentage of participants were female (66%), 60% had a Bachelor degree or higher, and 42% had a household income of $AUD2000 or more per week. Just under half (48%) had a chronic disease. A higher percentage of respondents used social media for more than 3 h per day (41%) compared to traditional media (24%). A large percentage strongly agreed (60%) or agreed (25%) that they would be willing to receive a COVID-19 vaccine if proven safe and effective. A large but slightly smaller percentage strongly agreed (58%) or agreed (24%) that they usually get vaccinated when recommended by a doctor. A much smaller percentage strongly agreed (4%) or agreed (12%) that they would receive a COVID-19 vaccine if NOT proven safe and effective. A small percentage were more willing to receive the COVID-19 vaccine compared to usual vaccines (8%); however, there was also a small percentage who were less willing to receive the COVID-19 vaccine compared to usual vaccines (11%).

There was a higher percentage of participants living in the state of Victoria at time 2 (35%) compared to time 1 (17%), *p* < 0.001, a higher percentage of adults over 65 at time 2 (27%) compared to time 1 (19%), *p* < 0.001, a lower percentage of females in the second (63%) compared to first survey (67%), *p* = 0.03, and a higher percentage of participants in the lowest household income category of less than $AUD1000 per week in the second (33%) compared to first survey (26%), *p* = 0.006. A lower percentage of participants had a chronic disease at time 2 (49%) compared with time 1 (53%), *p* = 0.03, and a lower percentage of participants used social media for more than 3 h per day at time 2 (36%) compared with time 1 (44%) *p* = 0.001.

Table 5 presents the percentage of participants who were willing to vaccinate against COVID-19 by time, demographics and media use. Most participants agreed that they would vaccinate against COVID-19 if the vaccine were proven safe and effective (85%).

The descriptive statistics demonstrated that the percentage of participants who were willing to vaccinate against COVID-19 decreased between survey 1 (86%) and survey 2 (82%). The decrease was greater for non-Victorians (from 86% to 81%) compared to Victorians (from 87% to 85%). A lower percentage of participants with technical studies/certificate or diploma were willing to vaccinate against COVID-19 (79%) compared to participants with a Bachelor degree (87%). A higher percentage of participants with a chronic disease were willing to vaccinate against COVID-19 (87%) compared to participants without a chronic disease (82%). A smaller percentage of people who listened or read less than 1 h of traditional media per day were willing to vaccinate (78%) compared to those who listened or read more than 3 h per day (89%).

Table 6 presents results of the multinomial logistic regression of willingness to vaccinate against COVID-19 by time, demographics, chronic disease status and media use. There were no significant associations between willingness to vaccinate and living location, age, chronic disease, or social media use. Participants were less likely to be willing to receive a COVID-19 vaccination in survey 2 compared with 1. The likelihood of agreeing and being unsure they would vaccinate compared to disagreeing was 0.45 and 0.52 times lower in survey 2 compared to survey 1. There was no significant difference between females and males on agreeing they would vaccinate; however, women were 1.89 times more likely to be unsure they would vaccinate compared with men. There was no significant difference between participants with a year 12 education or below compared to participants with a bachelor’s degree; however, those who had a certificate or diploma were 0.54 times less likely to agree they would vaccinate compared to those with a bachelor’s degree. Participants who listened or read less than 1 h of traditional media per day were 0.24 times less likely to agree and 0.39 times less likely to be unsure they would vaccinate compared to those who listened or read more than 3 h per day.

The secondary multinominal logistic regression analysis of usual willingness to vaccinate when recommended by a doctor by time, demographics, chronic disease, and media use is reported in Appendix A. Results demonstrated that adults aged 18–34 were 1.81 times more likely to be willing to vaccinate when recommended by their doctor compared to adults 65+ years. People with year 12 or below education were 0.62 times less likely to be willing compared to those with a Bachelor degree, people with a chronic disease were 1.88 times more likely to be willing compared to those without a chronic disease and people with <1 h traditional media use per day were 0.52 times less likely to be willing compared to those with >3 h of traditional media use.

## 4. Discussion

The main aims of this study were to determine whether willingness of Australian adults to vaccinate against COVID-19 changed from the beginning of the pandemic (April 2020) to four months later (August 2020) and to determine whether demographic factors, chronic disease status, traditional media use, and social media use are associated with willingness to vaccinate. Our results in the repeated sample demonstrate that willingness to vaccinate declined only slightly between survey 1 and survey 2 and this was not significant.

The percentage of participants willing to vaccinate against COVID-19 (85%) is above the estimated level of 60% required to achieve herd immunity given 80% vaccine efficacy [5], and is higher than the global average (74%) [25]. The Australian government is, however, aiming for a 95% vaccination rate, which is higher than this survey suggests we are likely to achieve. This demonstrates the need for vaccine promotion efforts to improve willingness to vaccinate against COVID-19. Interestingly, 19% of the sample differed in their willingness to vaccinate against COVID-19 compared to routine vaccinations. The proportion of these who were either more (8%) or less (11%) willing to vaccinate against COVID-19 was relatively similar. Further, the supplementary analysis found usual willingness to receive vaccines recommended by a doctor to be associated with different demographics (younger adults, Bachelor degree compared to year 12 education, and chronic disease) than willingness to vaccinate against COVID-19 (females, Bachelor degree compared to certificate or diploma education). This illustrates that there are other factors influencing people’s willingness to vaccinate against COVID-19 than usually choosing not to vaccinate when recommended by health professionals. This is in line with findings from a global survey conducted by the World Economic Forum that reported only 17% identified general opposition to vaccines as the reason for hesitancy to vaccinate against COVID-19 [25]. There are likely to be numerous reasons why people are hesitant to vaccinate against COVID-19 specifically. These might be related to the increasing discussion in the Australian media surrounding a concern about the safety and efficacy of a rapidly developed COVID-19 vaccine, related to conspiracy theories surrounding COVID-19, or related to a concern that one’s health condition (e.g., immunocompromised or pregnant) might mean that they cannot receive the vaccine until more testing is done. The World Economic Forum survey found that the most common reasons for hesitancy were worrying about side effects (56%) and doubts about vaccine effectiveness (29%) [25].

Whilst the repeated cross-sectional analysis in the full sample found a small, significant decrease in willingness to vaccinate between April and August 2020, the longitudinal analysis in the repeat sample found no association between time and willingness to vaccinate. Past research found changes in willingness to vaccinate over time in repeat cross sectional data [8,9,10] and the findings from the present study add to this by demonstrating that within person changes in willingness to vaccinate over time may not exist. However, the analysis in the repeat sample found that participants from the state of Victoria increased their willingness to vaccinate, causing a small interaction between survey (time 1, time 2) and living location (Victoria, other Australian state or territory) with marginal significance. It is therefore possible that willingness to vaccinate against COVID-19 may be influenced by lockdown restrictions impacting participants’ lives. People in strict lockdown conditions (due to increasing community transmissions) may be more likely to want to receive a vaccine to attenuate or stop the spread of COVID-19, and to avoid continued or future lockdown restrictions. This is also supported by the repeated cross-sectional study conducted in Italy where willingness to vaccinate against COVID-19 decreased over time as lockdown restrictions eased [11]. However, more research with additional time points is needed to determine whether willingness to vaccinate against COVID-19 is influenced by lockdown conditions.

Previous Australian surveys have demonstrated that willingness to receive a COVID-19 vaccination is positively associated with age [8,9]. Although this study found a slight increase in willingness to vaccinate with age, this was not significant. This study found no gender difference for agreeing compared to disagreeing to a COVID-19 vaccine; however, women were more likely than men to be unsure compared to disagreeing that they would receive a COVID-19 vaccine. This differs from past research showing that women are less likely to be willing to receive a COVID-19 vaccine [9,26]. As the gender difference seen in this study was due to women being more likely to be unsure about their willingness to vaccinate, this means women may be more likely to change their mind about receiving the vaccine and supports vaccine promotion efforts targeting women.

In line with previous Australian surveys, the present study demonstrated a significant association between willingness to vaccinate against COVID-19 and education level [8,9]. Similar to Rhodes and Hoq [9], this was due to lower willingness to vaccinate in participants with a certificate or diploma compared to participants with a bachelor’s degree. Participants with a Bachelor degree are more likely to have a higher health literacy or have specific training in health or research leading to a greater understanding of vaccine development, clinical trials, and the need for herd immunity to stop the spread of disease [27]. It is, however, not clear why the association exists for participants with a certificate or diploma and not a year 12 or lower education who also have a lower health literacy [27]. It is possible that people with no further education are working in positions with less secure employment [28]. Hence, they may be more likely to be financially impacted by COVID-19 restrictions, thereby increasing their inclination to vaccinate against COVID-19 as a means of reducing any further impacts of COVID-19 on their ability to secure an income. To reach those with a certificate or diploma, COVID-19 vaccine promotion efforts could target industries which employ a large number of certificate and diploma graduates including the automotive, building and construction, hospitality, fitness and beauty industries.

The lack of association between willingness to vaccinate against COVID-19 and chronic disease conflicts with past research that has found a positive association between distrust of childhood vaccinations and self-reported health [14], and the results of the supplementary analysis that people with a chronic disease are more likely to be willing to receive vaccines recommended by their doctor. It is possible that whilst people with a chronic disease may be more likely to have severe health problems if they are diagnosed with COVID-19 [16], they may not make the association between their current health status and how COVID-19 might exacerbate their symptoms. It is also possible that they are unsure if they will receive negative side effects after receiving a new vaccine.

Similarly, the lack of association between social media use and willingness to vaccinate against COVID-19 is not in line with past research on general vaccine hesitancy which found a high level of disinformation regarding the safety and efficacy of vaccinations on social media and the internet generally [17,18,19,20], and an association with using the internet to access health information and vaccine hesitancy [13,19]. It should be noted that the supplementary analysis also did not find an association between social media use and usual willingness to receive vaccines recommended by a doctor. The lack of association between social media and willingness to vaccinate may be because this study did not assess what participants spent their time doing on social media, therefore participants with high levels of social media use may be spending their time interacting with peers rather than consuming health or vaccine information through discussions or shared articles.

In contrast, this study found willingness to vaccinate was higher in participants who were high consumers of traditional media. This could be influenced by high traditional news consumers being older and more highly educated [29,30,31], although age was not significantly associated with willingness to vaccinate and these variables were controlled for in the analysis. The association between higher traditional media use and willingness to vaccinate may also be due to exposure to less biased news coverage of the negative effects of the pandemic and vaccine development and trials [25]. However, the supplementary analysis identified a smaller but also statistically significant positive association between traditional media use and usual willingness to receive vaccines recommended by a doctor. Therefore, the association between traditional media use and willingness to vaccinate against COVID-19 may be due to those with a lower use of traditional media having more skeptical views on vaccination in general. To reach low users of traditional media, COVID-19 vaccine promotion efforts could advertise through social media, which would also reach women who were more likely than men to be unsure about receiving a COVID-19 vaccine [32]. Low users of traditional media could also be reached at point of contact with the health care system. Discussing routine vaccination with physicians is positively associated with willingness to receive vaccinations [33]. This approach may also reach people with a chronic disease who were not more likely to be willing to vaccinate against COVID-19 despite them being more likely to have serious health problems if they are diagnosed with COVID-19.

### Strengths and Limitations

The strengths of this study include a large sample of Australians and two distributions of an identical survey at the beginning of the pandemic (April) and four months later (August). Another strength is the subset of participants who completed both surveys allowing for analyses of within person changes. However, a limitation of distributing the survey through existing networks and social media means we cannot take the sample to be representative of the general population. Compared to the Australian population, our sample was older (median age 55 compared to 37 years) and had a higher percentage of females (66% compared to 51%) [34]. A higher percentage of the sample had a bachelor’s degree compared to the Australian population (60% compared to 24%) [35]; however, the median household income ($1000–$1249) was similar to the average household income in Australia ($1062) [28]. Only 575 participants (24%) completed both surveys. Those who completed the follow up were slightly older (51%) compared to those who did not (46%). The other demographic characteristics and willingness to vaccinate against COVID-19 (85% compared to 86%) were similar. Despite this, it is possible that differential attrition may have affected the within person analyses. Recruitment methods between the two surveys were similar; however, the proportion of participants reached through the different recruitment avenues was not tracked. Further, demographics including location, age, gender, income, chronic disease status and social media use differed between the two surveys. Whilst demographics were controlled for, these differences may still have influenced the significant association between time and willingness to vaccinate in the repeated cross-sectional analysis. Lastly, at the time of the survey it was not clear if or when a vaccine and associated clinical safety data would become available in Australia. Once a vaccine is available, along with clinical trial findings, Australians’ willingness to vaccinate may change.

## 5. Conclusions

Results suggest that willingness to receive a COVID-19 vaccination is not changing over time within individuals. No associations were found between willingness to vaccinate and age; however, women were more likely to be unsure than unwilling to vaccinate against COVID-19 compared to males, and those with a certificate or diploma were less likely to be willing to vaccinate compared to those with a bachelor’s degree. There was no association between chronic disease or social media use; however, higher use of traditional media was positively associated with willingness to vaccinate. Promotion of a COVID-19 vaccine should consider targeting women, people with a certificate or diploma, people with a chronic disease and lower users of traditional media through social media, industries employing certificate and diploma graduates and the health care system. Future research in a larger longitudinal sample with additional time points is needed to determine whether willingness to vaccinate is changing, and associated with lockdown conditions. Such information will help to target groups with the greatest hesitancy to receive a COVID-19 vaccination to maximise vaccine uptake and minimise future COVID-19 infections in Australia.

## Figures and Tables

**Table 1 ijerph-18-00797-t001:** Descriptives of demographics and willingness to vaccinate against COVID-19 in follow up participants (*n* = 575).

Demographics and Willingness to Vaccinate	Variable Categories	Survey 1 Number (%)	Survey 2 Number (%)
Age	18–34	79 (13.7)	
	35–44	91 (15.8)	
	45–54	113 (19.7)	
	55–64	165 (28.7)	
	≥65	127 (22.1)	
Gender	Male	174 (30.4)	
	Female	399 (69.6)	
Education	Year 12 or below	64 (11.1)	
	Technical studies, certificate, diploma	146 (25.4)	
	Bachelor and above	365 (63.5)	
Household income	<$1000/week	136 (26.9)	
	$1000–<$2000/week	160 (31.7)	
	≥$2000/week	209 (41.4)	
Chronic disease status	With chronic disease	275 (47.8)	
	Without chronic disease	300 (52.2)	
Willingness to vaccinate against COVID-19 if proven safe and effective	Strongly agree	369 (64.2)	356 (61.9)
	Agree	132 (23.0)	131 (22.8)
	Neither agree nor disagree	46 (8.0)	58 (10.1)
	Disagree	18 (3.1)	22 (3.8)
	Strongly disagree	10 (1.7)	8 (1.4)
Location of residence	Victoria		109 (19.0)
	Other Australian state or territory		466 (81.0)

**Table 2 ijerph-18-00797-t002:** Descriptives of willingness to vaccinate against COVID-19 by time and location (*n* = 575).

Time	Location	Agree Number (%)	Neither Agree nor Disagree Number (%)	Disagree Number (%)
**Survey 1**	**Living location**			
	Victoria	93 (85.3)	8 (7.3)	8 (7.3)
	Other Australian state or territory	408 (87.6)	38 (8.2)	20 (4.3)
**Survey 2**	**Living location**			
	Victoria	97 (89.0)	9 (8.3)	3 (2.8)
	Other Australian state or territory	390 (83.7)	49 (10.5)	27 (5.8)

**Table 3 ijerph-18-00797-t003:** Linear Mixed Model of willingness to vaccinate against COVID-19 by time and location (*n* = 575).

Time and Location	Variable Categories	Agree OR (95% CI)	Neither Agree nor Disagree OR (95% CI)
Time	Survey 2	0.69 (0.37–1.26)	0.96 (0.47–1.96)
	Survey 1	Reference	Reference
Living location	Victorian resident	0.52 (0.21–130)	0.54 (0.17–1.67)
	Other resident	Reference	Reference
Time · Location of residence	Victorian resident	4.15 (0.92–18.66) ^a^	3.13 (0.52–18.66)
	Other resident	Reference	Reference

Reference category = Agree. Analysis controlled for Education due to its association with willingness to vaccinate in this sample. ^a^
*p* = 0.06. Note. Multinominal distribution with logit link.

**Table 4 ijerph-18-00797-t004:** Demographics and willingness to vaccinate against COVID-19 for survey 1 and 2 respondents (*n* = 2343).

Demographics and Willingness to Vaccinate	Variable Categories	Survey 1 Number (%)	Survey 2 Number (%)	Combined (%)	χ^2^
Overall		1512	831	2343	
Living Location	Victoria	262 (17.3)	289 (34.8)	551 (23.5)	90.78 ***
	Other Australian state or territory	1250 (82.7)	542 (65.2)	1792 (76.5)	
Age ^a^ (years)	18–34	266 (17.6)	99 (11.9)	365 (15.6)	35.72 ***
	35–44	260 (17.2)	107 (12.9)	367 (15.7)	
	45–54	284 (18.8)	154 (18.6)	438 (18.7)	
	55–64	419 (27.7)	226 (29.3)	509 (28.3)	
	≥65	283 (18.7)	226 (27.3)	509 (21.7)	
Gender ^a^	Male	491 (32.6)	306 (37.1)	797 (34.2)	4.68 *
	Female	1013 (67.4)	519 (62.9)	1532 (65.8)	
Education attained	Year 12 or below	231 (15.3)	141 (17.0)	372 (15.9)	1.25
	Technical studies, Certificate, Diploma	368 (24.3)	203 (24.4)	571 (24.4)	
	Bachelor degree or above	913 (60.4)	487 (58.6)	1400 (59.8)	
Household Income (AUD) ^a^	<$1000/week	343 (26.4)	229 (33.0)	572 (28.7)	10.08 **
	$1000–<$2000/week	388 (29.8)	194 (28.0)	582 (29.2)	
	≥$2000/week	570 (43.8)	270 (39.0)	840 (42.1)	
Chronic disease status	With chronic disease	703 (46.5)	425 (51.1)	1128 (48.1)	4.64 *
	Without chronic disease	809 (53.5)	406 (48.9)	1215 (51.9)	
Use of Social media	<1 h per day	290 (19.2)	177 (21.3)	467 (19.9)	13.74 **
	1–2 h per day	552 (36.5)	351 (42.2)	903 (38.5)	
	≥3 h per day	670 (44.3)	303 (36.5)	973 (41.5)	
Use of Traditional media	<1 h per day	549 (36.3)	297 (35.7)	846 (36.1)	0.69
	1–2 h per day	604 (39.9)	324 (39.0)	928 (39.6)	
	≥3 h per day	359 (23.7)	210 (25.3)	569 (24.3)	
Willingness to vaccinate against COVID-19 if proven safe and effective	Strongly agree	917 (60.6)	472 (56.8)	1389 (59.3)	18.06 **
	Agree	384 (25.4)	207 (24.9)	591 (25.2)	
	Neither agree nor disagree	130 (8.6)	74 (8.9)	204 (8.7)	
	Disagree	42 (2.8)	29 (3.5)	71 (3.0)	
	Strongly disagree	39 (2.6)	49 (5.9)	88 (3.8)	
Willingness to vaccinate against COVID-19 if NOT proven safe and effective	Strongly agree	45 (3.0)	41 (4.9)	86 (3.7)	12.00 *
	Agree	187 (12.4)	89 (10.7)	276 (11.8)	
	Neither agree nor disagree	391 (25.9)	226 (27.2)	617 (26.3)	
	Disagree	549 (36.3)	268 (32.3)	817 (34.9)	
	Strongly disagree	340 (22.5)	207 (24.9)	547 (23.3)	
Usually vaccinate when recommended by doctor	Strongly agree	855 (56.5)	509 (61.3)	1364 (58.2)	11.28 *
	Agree	388 (25.7)	178 (21.4)	566 (24.4)	
	Neither agree nor disagree	124 (8.2)	58 (7.0)	182 (7.8)	
	Disagree	85 (5.6)	39 (4.7)	124 (5.3)	
	Strongly disagree	60 (4.0)	47 (5.7)	107 (4.6)	
Willingness to vaccinate against COVID-19 compared to usual vaccination	More likely	100 (6.6)	78 (9.4)	178 (7.6)	11.26 **
	Equally likely	1227 (81.2)	680 (81.8)	1907 (81.4)	
	Less likely	185 (12.2)	78 (8.8)	178 (11.0)	

^a^ Missing. Survey 1: Age = 1, Gender = 8, Income = 211. Survey 2: Age = 2, Gender = 6, Income = 138. Note: Data from participants who completed both surveys were excluded at Survey 2 (*n* = 575). *** *p* < 0.001, ** *p* < 0.01, * *p* < 0.05.

**Table 5 ijerph-18-00797-t005:** Willingness to vaccinate against COVID-19 presented by time, demographics, chronic disease status and media use (*n* = 2328).

Demographics and Time	Variable Categories	Agree Number (%)	Neither Agree nor Disagree Number (%)	Disagree Number (%)
Overall		1972 (84.7)	204 (8.8)	152 (6.5)
Time	Survey 1	1296 (86.2)	130 (8.6)	78 (5.2)
	Survey 2	676 (82.0)	74 (9.0)	74 (9.0)
Living Location	Victoria	468 (85.7)	46 (8.4)	32 (5.9)
	Other Australian state or territory	1504 (84.4)	158 (8.9)	120 (6.7)
Time · Location	**Survey 1**			
	Victoria	225 (86.9)	21 (8.1)	13 (5.0)
	Other Australian state or territory	1071 (86.0)	109 (8.8)	65 (5.2)
	**Survey 2**			
	Victoria	243 (84.7)	25 (8.7)	19 (6.6)
	Other Australian state or territory	433 (80.6)	49 (9.1)	55 (10.2)
Age (years)	18–34	308 (85.3)	30 (8.3)	23 (6.4)
	35–44	293 (80.7)	37 (10.2)	33 (9.1)
	45–54	366 (83.8)	39 (8.9)	32 (7.3)
	55–64	563 (85.4)	56 (8.5)	40 (6.1)
	≥65	442 (87.0)	42 (8.3)	24 (4.7)
Gender	Male	674 (84.7)	58 (7.3)	64 (8.0)
	Female	1298 (84.7)	146 (9.5)	88 (5.7)
Education attained	Year 12 or below	310 (84.5)	37 (10.1)	20 (5.4)
	Technical studies, Certificate, Diploma	449 (79.3)	65 (11.5)	52 (9.2)
	Bachelor degree or above	1213 (87.0)	102 (7.3)	80 (5.7)
Chronic disease status	With chronic disease	979 (87.6)	76 (6.8)	63 (5.6)
	Without chronic disease	993 (82.1)	128 (10.6)	89 (7.4)
Use of social media	<1 h per day	384 (82.4)	44 (9.4)	38 (8.2)
	1–2 h per day	761 (84.7)	85 (9.5)	52 (5.8)
	≥3 h per day	827 (85.8)	75 (7.8)	62 (6.4)
Use of traditional media	<1 h per day	653 (78.1)	91 (10.9)	92 (11.0)
	1–2 h per day	814 (88.0)	69 (7.5)	42 (4.5)
	≥3 h per day	505 (89.1)	44 (7.8)	18 (3.2)

Note. Statement to assess willingness to vaccinate against COVID-19: ‘If a new vaccine for COVID-19 was released that was proven to be safe and effective, I would get vaccinated immediately’.

**Table 6 ijerph-18-00797-t006:** Multinomial logistic regression of willingness to vaccinate against COVID-19 by time, demographics, chronic disease status and media use (*n* = 2328).

Demographics and Time	Variable Categories	Agree OR (95% CI)	Neither Agree nor Disagree OR (95% CI)
Time	Survey 2	0.46 (0.31–0.67) ***	0.52 (0.32–0.86) *
	Survey 1	1.00	1.00
Living Location	Victoria	0.98 (0.52–1.82)	0.95 (0.44–2.04)
	Other Australian state or territory	1.00	1.00
Time · Location	Victoria	1.57 (0.68–3.63)	1.53 (0.54–4.36)
	Other Australian state or territory	1.00	1.00
Age (years)	18–34	0.99 (0.53–1.84)	0.84 (0.38–1.83)
	35–44	0.67 (0.38–1.20)	0.77 (0.37–1.58)
	45–54	0.78 (0.45–1.38)	0.79 (0.39–1.60)
	55–64	0.84 (0.49–1.43)	0.85 (0.44–1.64)
	≥65	1.00	1.00
Gender	Female	1.33 (0.94–1.89)	1.89 (1.20–2.97) **
	Male	1.00	1.00
Education	Year 12 or below	1.00 (0.59–1.68)	1.58 (0.84–2.99)
	Technical studies, Certificate, Diploma	0.54 (0.37–0.79) **	1.00 (0.62–1.61)
	Bachelor degree or above	1.00	1.00
Chronic disease status	With chronic disease	1.39 (0.98–1.97)	0.82 (0.53–1.27)
	Without chronic disease	1.00	1.00
Social media use	<1 h	1.03 (0.66–1.62)	1.23 (0.69–2.19)
	1–2 h	1.32 (0.88–1.96)	1.63 (0.99–2.69)
	>3 h	1.00	1.00
Traditional media use	<1 h	0.24 (0.14–0.42) ***	0.39 (0.20–0.76) **
	1–2 h	0.64 (0.36–1.14)	0.64 (0.32–1.27)
	>3 h	1.00	1.00

Reference category = Disagree. *** *p* < 0.001, ** *p* < 0.01, * *p* < 0.05. Note. Income was not included in the model due to a high number of missing values (*n* = 349). A bivariate multinomial logistic regression revealed no association between income and willingness to vaccinate *p* = 0.38.

## Data Availability

The data presented in this study are available on request from the corresponding author. The data are not publicly available due to ethical considerations.

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
