# Peer review of "As the Pandemic Progresses, How Does Willingness to Vaccinate against COVID-19 Evolve?"

_ijerph, 2021, doi:10.3390/ijerph18020797_

Round 1
Reviewer 1 Report
An interesting study, but fundamentally the authors are comparing two different samples and I believe the longitudinal sample should be those focused on.
- For the citation of the other Australian paper, the figures are incorrect. The 85% should be 86% and the 10%, should be 9%. There is also a follow-up to those findings recently accepted and will be published shortly in Lancet Infectious Diseases which shows data for June and July 2020.
- Where does the hypothesis about traditional or social media and willingness to vaccinate emerge from? It may be better to remove the first sentence of this paragraph as this comes from nowhere.
- It would be interesting to know what proportion of the sample came from each of the recruitment sources at both time points – was it mainly email lists or social media?
- The age categories used include a wide range of age groups. Do the authors expect that 18 year olds will have the same attitudes as 44 year olds?
- The fundamental problem with the methodology is that the same sample has not been compared as the main outcome of the study. There could be a number of confounding variables which contributed to the change in willingness to vaccinate across the two time points because it was not the same individuals and as you show in table 1, there were a number of differences between sample 1 and sample 2. Could the authors state if there are significant differences in the variables across the time points?
- Additional analysis could look at the social media/traditional media use by age group, but suggesting that the age groups be collapsed further so it would be more sensitive to variations across age groups.
- The numbers presented do not all match those in table 1 – please check these carefully and the rounding of decimal places in the text.
- Second paragraph of the discussion – to note, the government want to achieve 95% uptake of the vaccine.
- So what are the implications of your findings? Some more discussion about how these findings can be used to support COVID-19 vaccine strategies would be advantageous.
- Line 323 – herd immunity, not ‘heard’ immunity.
Author Response
Thank you for your insightful comments on our manuscript which we believe has further improved the manuscript. We have responded to your comments below alongside related extracts from the paper. Changes to the manuscript text are highlighted using track changes. Changes to the tables are highlighted in red text.
Comment 1: An interesting study, but fundamentally the authors are comparing two different samples and I believe the longitudinal sample should be those focused on.
Author response: We agree there is benefit to focusing on the longitudinal sample and have brought the longitudinal analysis to the forefront of the results (Tables 1-3) and updated the abstract and discussion to focus on these longitudinal comparisons of willingness to vaccinate against COVID over time instead of the repeated cross-sectional analysis.
Abstract, page 1, lines 22-19:
Original: ‘Survey data was collected in April and August 2020 from a total of 2343 Australian adults. A multinominal logistic regression was conducted to determine whether willingness to vaccinate changed between the beginning of the pandemic (April 2020) to four months later (August 2020), and to determine whether willingness to vaccinate was associated with demographics, chronic disease, or media use. Willingness to vaccinate was high but decreased between April (86%) and August (82%), p<.001.
Revised: ‘Survey data were collected in April and August 2020 from a total of 2343 Australian adults. A quarter (n=575, 24%) completed both surveys. A generalized linear mixed model analysis was conducted to determine whether willingness to vaccinate changed in the repeated sample, and a multinominal logistic regression was conducted in all participants to determine whether willingness to vaccinate was associated with demographics, chronic disease, or media use. Willingness to vaccinate slightly decreased between April (87%) and August (85%) but this was not significant.
Discussion page 11, lines 362-394:
Original: ‘The significant decrease in willingness to vaccinate between survey 1 and survey 2 is in line with findings from two studies, conducted independently from one another in Australia at different times points in different samples, which found willingness to vaccinate decreased between April and June 2020 [8, 9]. Our findings add to this by demonstrating a decrease between April and August 2020 in an identical survey distributed twice with consistent methods and question wording. This drop in willingness to vaccinate may be due to decreasing community transition of COVID-19 and reduced lockdown laws in all Australian states except Victoria. There has also been increasing discussion in the Australian media surrounding COVID-19 vaccination trials and a growing public concern about the safety and efficacy of a rapidly developed COVID-19 vaccine [24]. The supplementary analysis in the subsample of participants who completed both surveys also showed a decreasing trend in willingness to vaccinate, however this was not significant. A larger longitudinal survey in repeat participants is required to determine whether within person time differences in willingness to vaccinate against COVID-19 exist.
Despite the decrease in willingness to vaccinate being much lower in Victorians who were in lockdown during both survey time points, there was no significant interaction between time and living location (Victoria, other Australian state or territory). However, the within person supplementary analysis found that participants from the state of Victoria increased their willingness to vaccinate, demonstrating an interaction between survey (time 1, time 2) and living location (Victoria, other Australian state or territory) with marginal significance. It is therefore possible that the decrease in willingness to vaccinate observed between survey 1 and 2 may be influenced by lockdown restrictions impacting participants’ lives. People in strict lockdown conditions (due to increasing community transmissions) may be more likely to want to receive a vaccine to attenuate or stop the spread of COVID-19, and to avoid continued or future lockdown restrictions. This is also supported by the repeated cross-sectional study conducted in Italy where willingness to vaccinate against COVID-19 decreased over time as lockdown restrictions eased [10]. However, more research with additional time points is needed to determine whether the decreasing trend is due to changing lockdown conditions.’
Revised: ‘Whilst the repeated cross-sectional analysis in the full sample found a small, significant decrease in willingness to vaccinate between April and August 2020, the longitudinal analysis in the repeated sample found no association between time and willingness to vaccinate. Past research found changes in willingness to vaccinate over time in repeat cross sectional data [8, 9] and the findings from the present study add to this by demonstrating that within person changes in willingness to vaccinate over time may not exist. However, the analysis in the repeat sample found that participants from the state of Victoria increased their willingness to vaccinate, causing a small interaction between survey (time 1, time 2) and living location (Victoria, other Australian state or territory) with marginal significance. It is therefore possible that willingness to vaccinate against COVID-19 may be influenced by lockdown restrictions impacting participants’ lives. People in strict lockdown conditions (due to increasing community transmissions) may be more likely to want to receive a vaccine to attenuate or stop the spread of COVID-19, and to avoid continued or future lockdown restrictions. This is also supported by the repeated cross-sectional study conducted in Italy where willingness to vaccinate against COVID-19 decreased over time as lockdown restrictions eased [10]. However, more research with additional time points is needed to determine whether willingness to vaccinate against COVID-19 is influenced by lockdown conditions.’
Comment 2: For the citation of the other Australian paper, the figures are incorrect. The 85% should be 86% and the 10%, should be 9%. There is also a follow-up to those findings recently accepted and will be published shortly in Lancet Infectious Diseases which shows data for June and July 2020.
Author response: Thank you for picking this up. We have updated these figures to read 86% and 9% (Introduction, page 2). We have now also referred to the recently published follow up findings of this study.
Introduction, page 2, lines56-63:
Original: A survey conducted during the initial wave of the pandemic in Australia (April 2020) found that 86% of Australians were willing to receive a COVID-19 vaccine (another 9% were unsure, and 5% were unwilling) [8].
Revised: A survey conducted during the initial wave of the pandemic in Australia (April 2020) found that 86% of Australians were willing to receive a COVID-19 vaccine (another 9% were unsure, and 5% were unwilling) [8]. However, some evidence suggests that willingness to vaccinate against COVID-19 is changing in Australia [9]. The survey was conducted again in June and July 2020 when COVID-19 transmission had slowed and social restrictions were easing. Results showed a slight increase in willingness to vaccinate where 87% of June participants and 90% of July participants were willing to receive a COVID-19 vaccine. [9].
Comment 3: Where does the hypothesis about traditional or social media and willingness to vaccinate emerge from? It may be better to remove the first sentence of this paragraph as this comes from nowhere.
Author response: We agree and have removed the strongly worded introductory statement that media use may be associated with willingness to vaccinate against COVID (Introduction, page 2, lines 87-88). Now the paragraph begins by discussing how people who frequently access the news may be more informed about COVID-19 and vaccine development and how this may in turn affect their willingness to vaccinate against COVID-19. The paragraph ends with a discussion of how this may differ between social media and traditional news media as misinformation is rife on social media and social media use is positively associated with negative attitudes towards routine vaccines.
Comment 4: It would be interesting to know what proportion of the sample came from each of the recruitment sources at both time points – was it mainly email lists or social media?
Author response: Unfortunately, we did not track the proportion of participants reached through the different recruitment avenues. However, considering we used the same recruitment methods in both surveys it is unlikely that this would differ significantly. We have now included a comment about this in the strengths and limitations section:
Strengths and limitations, page 13, lines 469-471:
Original: ‘Nevertheless, recruitment methods and demographics between the two surveys were similar.’
Revised: ‘Recruitment methods between the two surveys were similar, however the proportion of participants reached through the different recruitment avenues was not tracked.’
Comment 5: The age categories used include a wide range of age groups. Do the authors expect that 18 year olds will have the same attitudes as 44 year olds?
Author response: We agree that the categories of the age variable are wide. We have therefore changed the age variable to include 5 instead of 3 categories in all tables reporting descriptive statistics and analyses: 1) 18-34, 2) 35-44, 3) 45-54, 4) 55-64, 5) ≥65 (see tables 1, 4, 5, 6). This did not change the outcomes of any analysis.
Comment 6: The fundamental problem with the methodology is that the same sample has not been compared as the main outcome of the study. There could be a number of confounding variables which contributed to the change in willingness to vaccinate across the two time points because it was not the same individuals and as you show in table 1, there were a number of differences between sample 1 and sample 2. Could the authors state if there are significant differences in the variables across the time points?
Author response: We agree that this is a limitation to the repeated cross-sectional analysis in the full sample looking at changes in willingness to vaccinate against COVID-19 over time. We have therefore included the longitudinal analysis of the participants who completed both surveys as the main analysis testing changes in willingness to vaccinate against COVID-19 over time. We have also statistically compared the demographics across the two surveys. There were differences in all demographics except education and traditional media use (See table 4). These variables were controlled in the full sample analyses.
Results, page 7, lines 270-278:
‘There was a higher percentage of participants living in the state of Victoria at time 2 (35%) compared to time 1 (17%), p<.001, a higher percentage of adults over 65 at time 2 (27%) compared to time 1 (19%), p<.001, a lower percentage of females in the second (63%) compared to first survey (67%), p=.03, and a higher percentage of participants in the lowest household income category of less than $AUD1000 per week in the second (33%) compared to first survey (26%), p=.006. A lower percentage of participants had a chronic disease at time 2 (49%) compared with time 1 (53%), p=0.3, and a lower percentage of participants used social media for more than 3 hours per day at time 2 (36%) compared with time 1 (44%) p=.001.’
We have highlighted this limitation of the cross-sectional analysis in the discussion.
Strengths and limitations, page 13, lines 471-474:
‘Further, demographics including location, age, gender, income, chronic disease status and social media use differed between the two surveys. Whilst demographics were controlled for, these differences may still have influenced the significant association between time and willingness to vaccinate in the repeated cross-sectional analysis.’
Comment 7: Additional analysis could look at the social media/traditional media use by age group, but suggesting that the age groups be collapsed further so it would be more sensitive to variations across age groups.
Author response: Comparing media use by age group is beyond the scope of this study, as the manuscript already presents a substantial number of analyses, however we agree that the 3 age categories were too wide and have broken these down into 5 categories (see response to comment 5). This will better control for age when assessing the relationship between media use and willingness to vaccinate.
Comment 8: The numbers presented do not all match those in table 1 – please check these carefully and the rounding of decimal places in the text.
Author response: Thank you for picking these errors up. The percentages of participants responding to agree and strongly agree to the three questions have now been updated. The percentage of females in the sample has also been updated to read 36% instead of 35% (see Results, page 7, lines 259-266).
Comment 9: Second paragraph of the discussion – to note, the government want to achieve 95% uptake of the vaccine.
Author response: Thank you for bringing this to our attention. We have highlighted this in the discussion.
Discussion, page 10, lines 340-343:
‘The Australian government is however aiming for a 95% vaccination rate, which is higher than this survey suggests we are likely to achieve. This demonstrates the need for vaccine promotion efforts to improve willingness to vaccinate against COVID-19.’
Comment 10: So what are the implications of your findings? Some more discussion about how these findings can be used to support COVID-19 vaccine strategies would be advantageous.
Author response: We have expanded on the implication of our findings by further discussing how the groups with low willingness to vaccinate could be reached including promotion strategies that may be well suited to targeting these groups:
Discussion, page 12, lines 415-418:
‘To reach those with a certificate or diploma, COVID-19 vaccine promotion efforts could target industries which employ a large number of certificate and diploma graduates including automotive, building and construction, hospitality, fitness and beauty industries.’
Discussion, page 12, lines 447-454:
‘To reach low users of traditional media, COVID-19 vaccine promotion efforts could advertise through social media which would also reach women who were more likely than men to be unsure about receiving a COVID-19 vaccine [31]. Low users of traditional media could also be reached at point of contact with the health care system. Discussing routine vaccination with physicians is positively associated with willingness to receive vaccinations [32]. This approach may also reach people with a chronic disease who were not more likely to be willing to vaccinate against COVID-19 despite them being more likely to have serious health problems if they are diagnosed with COVID-19.’
Conclusions, page 13, lines 488-491:
‘Promotion of a COVID-19 vaccine should consider targeting women, people with a certificate or diploma, people with a chronic disease and lower users of traditional media through social media, industries employing certificate and diploma graduates and the health care system.’
Comment 11: Line 323 – herd immunity, not ‘heard’ immunity.
Author response: Thank you for picking up this typo. It has now been corrected to read ‘herd’ instead of ‘heard’ (discussion, page 11, line 409).
Reviewer 2 Report
Research topics are hot issues that people are currently concerned about.
The research content has very important social value. The research method is appropriate and the content is clearly expressed.
However, the current research still needs to complete the discussion section, such as supplementing the potential reasons why the population is unwilling to be vaccinated and analyzing the advantages and disadvantages of using existing vaccines.
Author Response
Thank you for your comments on our manuscript. We have responded to your comments below alongside related extracts from the paper. Changes to the manuscript are highlighted using track changes.
Comment 1: Research topics are hot issues that people are currently concerned about.
Author response: Thank you for your positive comment
Comment 2: The research content has very important social value. The research method is appropriate and the content is clearly expressed.
Author response: Thank you for your positive comments. We agree that this research has an important social value. Australia will soon be rolling out a COVID vaccination where the hurdle will be assuring enough Australians agree to receive the vaccine. Our research informs the targeting of vaccination promotion efforts.
Comment 3: However, the current research still needs to complete the discussion section, such as supplementing the potential reasons why the population is unwilling to be vaccinated and analyzing the advantages and disadvantages of using existing vaccines.
Author response: Thank you for your comment. We have further improved the discussion by expanding on our discussion of why some people may be unwilling to receive a COVID vaccination. As this was not the aim of our research and we do not have the data to inform reasons why people are unwilling to vaccinate against COVID-19, we have kept this brief. We have instead focused the discussion on what is associated with willingness to vaccinate against COVID-19 which directly relates to our findings.
Discussion, page 10, lines 351-361:
Original: ‘This is in line with findings from a global survey conducted by the World Economic Forum that reported only 17% identified general opposition to vaccines as the reason for hesitancy to vaccinate against COVID-19. In contrast, the most common reasons for hesitancy were worry about side effects (56%) and doubts about effectiveness (29%) [24].’
Revised: ‘This is in line with findings from a global survey conducted by the World Economic Forum that reported only 17% identified general opposition to vaccines as the reason for hesitancy to vaccinate against COVID-19 [24]. There are likely numerous reasons why people are hesitant to vaccinate against COVID-19 specifically. These might be related to the increasing discussion in the Australian media surrounding a concern about the safety and efficacy of a rapidly developed COVID-19 vaccine, related to conspiracy theories surrounding COVID_19, or related to a concern that one’s health condition (e.g. immunocompromised or pregnant) might mean that they cannot receive the vaccine until more testing is done. The world economic forum survey found that the most common reasons for hesitancy were worrying about side effects (56%) and doubts about vaccine effectiveness (29%) [24].’
Reviewer 3 Report
The authors have performed the same set of questions on Australian adult participants twice during the COVID-19 pandemic to evaluate to evaluate changes in perceptions of vaccination against COVID-19 vaccines as the pandemic and restrictions evolved as well factors that may influence the decision to vaccinate. There were a large number of participants in one survey although only 24% completed two surveys which is a significant limitation in the study and should be discussed in the limitations. However, the results are interesting and very topical which makes this paper of interest to readers. The incorporation of a state in lockdown compared to states not having such severe restrictions is interesting. I have the following suggestions;
- The fact that only 24% of patients answered 2 surveys should be discussed in the limitations and further details are required to determine whether there is evidence of participation bias for those who chose to answer the survey twice.
- Many of the results are statistically significant, given the large sample size, but in a practical sense quite small. This should be identified as a limitation in the discussion.
- Were any questions that may act as a calibrating variable used? For example, the willingness of individuals to take the influenza vaccine which could be compared to the COVID-19 vaccine results to work out how much of the results are COVID-19 specific. If not, this should be discussed as a limitation
- Minor points:
- Page 2, Line 47 - the sentence says 60% of people need to be vaccinated but does not describe which outcome this will achieve. Please elaborate.
- Please explain what Qualtrics is briefly for readers and add location it is based out of.
- Page 4 Line 182 - change "compliment" to "complement" Section 2.2 - a supplemental Table detailing the questions asked would aid the text provided regarding the content of the survey
- Results: Paragraph 2 - did the authors consider providing tests for significance in the comparisons as this will provide more information than just percentages provided.
- Page 7 line 261 0.06 is not statistically significant, consider changing wording of this sentence
- Page 8 line 287 change transition to transmission
Author Response
Thank you for your insightful comments on our manuscript which we believe has further improved the manuscript. We have responded to your comments below alongside related extracts from the paper. Changes to the manuscript text are highlighted using track changes. Changes to the tables are highlighted in red text.
Comment 1: The authors have performed the same set of questions on Australian adult participants twice during the COVID-19 pandemic to evaluate to evaluate changes in perceptions of vaccination against COVID-19 vaccines as the pandemic and restrictions evolved as well factors that may influence the decision to vaccinate. There were a large number of participants in one survey although only 24% completed two surveys which is a significant limitation in the study and should be discussed in the limitations. However, the results are interesting and very topical which makes this paper of interest to readers. The incorporation of a state in lockdown compared to states not having such severe restrictions is interesting. I have the following suggestions;
Author response: Thank you for your comment. We agree that a limitation of the study is the small percentage of participants who completed both surveys and have addressed this in response to your comments below (comment 2).
Comment 2: The fact that only 24% of patients answered 2 surveys should be discussed in the limitations and further details are required to determine whether there is evidence of participation bias for those who chose to answer the survey twice.
Author response: We have highlighted in the limitations section that only a small percentage of participants completed both surveys and discussed the likely impact of participation bias:
Strengths and limitations, page 12, lines 465-469:
Original: ‘Only 575 participants completed both surveys, thus, it is possible that differential attrition may have affected these comparisons.’
Revised: ‘Only 575 participants (24%) completed both surveys. Those who completed the follow up were slightly older (51%) compared to those who did not (46%). The other demographic characteristics and willingness to vaccinate against COVID-19 (85% compared to 86%) were similar. Despite this, it is possible that differential attrition may have affected the within person analyses.’
Comment 3: Many of the results are statistically significant, given the large sample size, but in a practical sense quite small. This should be identified as a limitation in the discussion.
Author response: We agree that some of the significant findings relate to small effects, however many of the significant associations between demographics and willingness to vaccinate relate to moderate effects. For example, those who had a certificate or diploma where 0.54 times less likely to agree they would vaccinate compared to those with a Bachelor degree (79% vs 87%). We have highlighted that the significant findings of changes over time only relate to small effects:
Discussion, page 11, lines 362-364:
‘Whilst the repeated cross-sectional analysis in the full sample found a small, significant decrease in willingness to vaccinate between April and August 2020, the longitudinal analysis in follow up participants found no association between time and willingness to vaccinate.’
Discussion, page 11, lines 382-386:
‘However, the within person analysis found that participants from the state of Victoria increased their willingness to vaccinate, demonstrating a small interaction between survey (time 1, time 2) and living location (Victoria, other Australian state or territory) with marginal significance.’
Comment 4: Were any questions that may act as a calibrating variable used? For example, the willingness of individuals to take the influenza vaccine which could be compared to the COVID-19 vaccine results to work out how much of the results are COVID-19 specific. If not, this should be discussed as a limitation
Author response: Thank you for this great suggestion. We have replicated the cross-sectional analyses investigating the associations between demographics and willingness to vaccinate for the outcome of usual willingness to receive vaccines recommended by a doctor (see supplementary file 1). This analysis revealed that people with no further education and lower users of traditional media were less likely to be willing to vaccinate whilst younger adults and people with a chronic disease were more likely to be willing to vaccinate. These findings are compared to the outcomes of the main analysis in the discussion section:
‘Discussion, page 10, lines 346-351:
‘Further, the supplementary analysis found usual willingness to receive vaccines recommended by a doctor to be associated with different demographics (younger adults, Bachelor degree compared to year 12 education, and chronic disease) than willingness to vaccinate against COVID-19 (females, Bachelor degree compared to certificate or diploma education). This illustrates that there are other factors influencing people’s willingness to vaccinate against COVID-19 than usually choosing not to vaccinate when recommended by health professionals.’
Discussion, page 12, lines 419-422:
‘The lack of association between willingness to vaccinate against COVID-19 and chronic disease conflicts with past research that has found a positive association between distrust in childhood vaccinations and self-reported health [13], and the results of the supplementary analysis that people with a chronic disease are more likely to be willing to receive vaccines recommended by their doctor.’
Discussion, page 12, lines 431-436:
‘It should be noted that the supplementary analysis also did not find an association between social media use and usual willingness to receive vaccines recommended by a doctor. The lack of association between social media and willingness to vaccinate may be as this study did not assess what participants spent their time doing on social media, therefore participants with high levels of social media use may be spending their time interacting with peers rather than consuming health or vaccine information through discussions or shared articles.’
Discussion, page 12, lines 433-447:
‘However, the supplementary analysis identified a smaller but also statistically significant positive association between traditional media use and usual willingness to receive vaccines recommended by a doctor. Therefore, the association between traditional media use and willingness to vaccinate against COVID-19 may be due to those with a lower use of traditional media having more skeptical views on vaccination in general.’’
Minor points:
Comment 5: Page 2, Line 47 - the sentence says 60% of people need to be vaccinated but does not describe which outcome this will achieve. Please elaborate.
Author response: We have now explained that the goal outcome is to achieve herd immunity:
Introduction, page 2, lines 51-52:
Original: ‘For a COVID-19 vaccine with 80% efficacy, it is estimated that at least 60% of the population will need to receive the vaccination [5].’
Revised: For a COVID-19 vaccine with 80% efficacy, it is estimated that at least 60% of the population will need to receive the vaccination to achieve herd immunity [5].
Comment 6: Please explain what Qualtrics is briefly for readers and add location it is based out of.
Author response: We have added in an explanation Qualtrics.
Materials and Methods, page 3, lines 110-112:
Original: ‘Participants were recruited to complete an online survey hosted on Qualtrics [20].’
Revised: ‘Participants were recruited to complete an online survey on Qualtrics. Qualtrics is an online platform which hosts secure surveys and is based in Seattle, Washington and Provo, Utah. Participants are directed to the survey through a direct link [20].’
Comment 7: Page 4 Line 182 - change "compliment" to "complement" Section 2.2 - a supplemental Table detailing the questions asked would aid the text provided regarding the content of the survey
Author response: Thank you for picking up this typo. We have updated this to read ‘complement.’
Comment 8: Results: Paragraph 2 - did the authors consider providing tests for significance in the comparisons as this will provide more information than just percentages provided.
Author response: We have now included significance tests in Table 4 to compare the two surveys.
Results, page 7, lines 270-278:
‘There was a higher percentage of participants living in the state of Victoria at time 2 (35%) compared to time 1 (17%), p<.001, a higher percentage of adults over 65 at time 2 (27%) compared to time 1 (19%), p<.001, a slightly lower percentage of females in the second (63%) compared to first survey (67%), p=.03, and a higher percentage of participants in the lowest household income category of less than $AUD1000 per week in the second (33%) compared to first survey (26%), p=.006. A lower percentage of participants had a chronic disease at time 2 (49%) compared with time 1 (53%), p=0.03, and a lower percentage of participants used social media for more than 3 hours per day at time 2 (36%) compared with time 1 (44%) p=.001.’
We have also included a discussion of these differences in the limitations section:
Strengths and limitations, page 13, lines 471-474:
‘Further, demographics including location, age, gender, income, chronic disease status and social media use differed between the two surveys. Whilst demographics were controlled for, these differences may still have influenced the significant association between time and willingness to vaccinate in the cross-sectional analysis.’
Comment 9: Page 7 line 261 0.06 is not statistically significant, consider changing wording of this sentence
Author response: We have edited this sentence to improve clarity:
Results, page 5:
Original: ‘Although it did not meet criteria for statistical significance (p = .06), there was a marginally statistically significant interaction between location and time, (p=0.06) where Victorians were more likely to increase their willingness to vaccinate between time 1 and 2.’
Revised: ‘Although it did not meet criteria for statistical significance (p=.06), there was some indication that Victorians were more likely to increase their willingness to vaccinate between time 1 and 2.’
Comment 10: Page 8 line 287 change transition to transmission
Author response: Thank you for picking up this typo. This sentence has been updated in response to other reviewer comments and no longer includes the word ‘transmission.’ We have ensured ‘transmission’ is spelt correctly elsewhere in the manuscript.
Section 2.2 - a supplemental Table detailing the questions asked would aid the text provided regarding the content of the survey
Author response: We have added a supplementary file (Supplementary file 1) detailing all questions asked in the survey that were used in this manuscript.
Round 2
Reviewer 1 Report
Thank you to the authors for attending to my comments. I am now satisfied with this manuscript, with one minor addition needed to some recent data cited.
Need to include a reference for the most recent Australian data cited (https://www.thelancet.com/journals/laninf/article/PIIS1473-3099(20)30926-9/fulltext#%20)
Author Response
Comment 1.
Need to include a reference for the most recent Australian data cited (https://www.thelancet.com/journals/laninf/article/PIIS1473-3099(20)30926-9/fulltext#%20)
Author response.
We thank Reviewer 1 for highlighting this oversight. We have now included the reference as requested on L63 of the manuscript as follows:
Results showed a slight increase in willingness to vaccinate where 87% of June participants and 90% of July participants were willing to receive a COVID-19 vaccine [10].
We have added this paper to the reference list and updated citation numbers of the following references accordingly.
Reference:
Dodd RH, Pickles K, Nickel B, Cvejic E, Ayre J, Batcup C, Bonner C, Copp T, Cornell S, Dakin T, et al.: Concerns and motivations about COVID-19 vaccination. Lancet Infect. Dis. 2020.
Reviewer 3 Report
The authors have thoroughly addressed all of the suggestions. I commend them on their paper.
Author Response
Comment 1:
The authors have thoroughly addressed all of the suggestions. I commend them on their paper.
Author response:
Thank you for reviewing our manuscript and for your suggestions which helped to improve the paper.